# Ecological Aspects of the Phlebotominae Fauna (Diptera: Psychodidae) among Forest Fragments and Built Areas in an Endemic Area of American Visceral Leishmaniasis in João Pessoa, Paraíba, Brazil

**DOI:** 10.3390/insects13121156

**Published:** 2022-12-14

**Authors:** Bruna Queiroz da Silva, Margarete Martins dos Santos Afonso, Lucas José Macêdo Freire, Antônio Luís Ferreira de Santana, Alessandre Pereira-Colavite, Elizabeth Ferreira Rangel

**Affiliations:** 1Laboratório Interdisciplinar de Vigilância em Diptera e Hemmiptera, Instituto Oswaldo Cruz/FIOCRUZ-RJ, Rio de Janeiro 21040-900, RJ, Brazil; 2Laboratório de Entomologia, Departamento de Sistemática e Ecologia, CCEN, Universidade Federal da Paraíba, João Pessoa 58052-900, PB, Brazil

**Keywords:** sandfly, richness, diversity, urban area, urban forest

## Abstract

**Simple Summary:**

Sand flies are small dipterans of medical importance, as some species are vectors of American visceral leishmaniasis disease (AVL). To determine whether the community of these vectors in forest areas and built areas are influenced by environmental conditions in the city of João Pessoa (state of Paraíba, Brazil), collections were carried out between March 2019 and July 2021. The results showed that forested areas have more non-ALV vector species; however, in the built areas, the predominant species was *Lutzomyia longipalpis*, a known vector of AVL. This suggests that *L. longipalpis* adapts better to anthropized areas than to more preserved areas. Thus, health authorities must intensify AVL control programs in the locality.

**Abstract:**

Sand flies are dipterans of medical importance, as some species are vectors of American visceral leishmaniasis (AVL). The municipality of João Pessoa (Paraíba, northeastern Brazil), is an endemic region for AVL, having high rates of human and canine cases. The main objective was to evaluate the sand fly fauna among forest fragments and built areas, and its relationship with environmental conditions. HP light traps were placed in the studied areas from March 2019 to July 2021. A total of 2141 specimens of phlebotomines were captured, comprising nine genera and ten species. Temperature and humidity were significant and positive only in built areas. The diversity composition among forest fragments and built areas was different and the AVL vector, *Lutzomyia longipalpis*, was the most prevalent species in built areas. The study showed that the built areas present differences in their richness and diversity of sand flies in relation to forest fragments, concluding that the conservation of forest areas, even if urban fragments, favors the diversity of phlebotomine species.

## 1. Introduction

Sand flies (Diptera: Psychodidae: Phlebotominae) are dipterans of medical importance since some species are vectors of *Leishmania* spp., *Bartonella bacilliformis*, and some arboviruses. Cosmopolitans, 1047 species are currently known (1014 living and 31 fossils), of which 277 are recorded in Brazil, where about 10% are related as vectors of leishmaniasis [1,2,3,4,5]. Leishmaniasis is a group of diseases caused by parasitic protozoa of the genus *Leishmania*, which cause clinical manifestations that can compromise the mucosa, skin, and viscera. The disease occurs in two forms, visceral leishmaniasis and cutaneous leishmaniasis (cutaneous, mucosal, and mucocutaneous) [6,7,8].

American visceral leishmaniasis (AVL) is a parasitic disease caused by the protozoan *Leishmania* (*L*.) *infantum chagasi*, and without proper diagnosis and treatment, it is fatal in 95% of cases [9,10]. About 1 billion people live in endemic regions for leishmaniasis and it is estimated that 30,000 new cases of the disease occur each year worldwide [6]. Poverty and socio–environmental conditions are identified as the main factors for the prevalence of AVL, being initially registered in rural or peri-urban areas [10,11]. However, the disease is increasing in large urban areas, and this epidemiological change has been associated with migration, environmental degradation, and climate change [12,13,14,15,16].

The epidemiology of a disease is a complex process where several variables are important in understanding the associated agents. In other Diptera, for example, community diversity can affect disease transmission dynamics, as the loss of biodiversity almost always causes an increase in vector species, which contributes to its prevalence [17,18,19]. Environmental changes impact the vector and host populations [20]. Furthermore, even slight temperature fluctuations can act to accelerate the developmental cycle of *Leishmania* in sand flies, which can potentiate the transmission of the protozoan [21,22,23,24,25].

Knowledge on the diversity of the sand fly fauna and its ecological aspects contribute to understanding AVL transmission dynamics [8,26,27]. In Brazil, the main vector of AVL is *Lutzomyia longipalpis* (Lutz and Neiva, 1912), and the vectorial importance of *Lutzomyia cruzi* (Mangabeira, 1938) in the Central-West region, and *Migonemyia migonei* (França, 1920) in the state of Pernambuco, northeastern Brazil, has already been recognized [11,28,29,30,31,32,33,34,35,36].

Brazil recorded 97% of AVL cases in the Americas in 2020, with the northeast region accounting for 49% of cases [37]. The city of João Pessoa, located in the state of Paraíba, northeast Brazil, was considered an area of intense transmission for AVL, having recorded 113 cases of the disease in the last 10 years, with a detection rate of 13.68 cases per 100,000 inhabitants [10,38,39].

Comparative studies among forested and human-affected areas in the Brazilian northeast indicate a correlation between diversity decrease in phlebotomine populations and abundance increase of vector species due to the anthropization process, a fact possibly associated with the adaptive capacity of some species such as *Lu. longipalpis* and *Nyssomyia whitmani* (Antunes and Coutinho, 1939) [40,41,42].

Studies on sand fly fauna are scarce in João Pessoa, despite being an endemic area of AVL with constant notification of human and canine cases [43,44]. In this way, the objective of this study was to survey the sand fly species in João Pessoa, in forested and built areas, as well as to compare the diversity and abundance among these areas and their relationship with environmental conditions.

## 2. Materials and Methods

### 2.1. Study Area

The study was carried out in the city of João Pessoa (7°06′54″ S, 34°51′47″ W), capital of the Paraiba state (Figure 1A). This municipality has an area of 210.044 km^2^ housing 825,796 inhabitants (3421.28 inhab./km^2^), considered the most populous in the state. According the Brazilian Institute of Geography and Statistics, 70.8% of its homes have sewage collection and 78.4% have arborized streets [45]. João Pessoa has a humid tropical climate according Köppen [46], with a high gradient of precipitation and average temperatures ranging from 21 °C to 30 °C. The annual rainfall index is 1869 mm/year and maximum precipitation occurs between May and July [47]. João Pessoa is located in the domain of the Atlantic Forest biome, composed of ombrophilous and semideciduous forests, tabuleiro forests, mangroves, and restingas [48]. The main economic activities are commerce, service, and tourism, the latter with an annual visitation of around 1.6 million people [45,49]. João Pessoa has many forest fragments, including 8 urban parks, 19 conservation units and 6 forest remnants close to the built areas [48].

The captures of the phlebotominae were carried out in the households of the edified area and in the two of the largest forest fragments of the João Pessoa: the Benjamin Maranhão Botanical Garden (hereafter BMBG) and the Mata de Jacarapé State Park (hereafter MJSP) (380 ha) [48].

The BMBG (07°08′08″ S, 34°51′37.3″ W) is in the urban matrix. It is a typical Atlantic rainforest with temperatures ranging from 21 °C to 30 °C and relative humidity between 73% and 82%. A total of 513 plant species, 55 fungi, 331 invertebrates (including bees, butterflies, dung beetles, scorpions, and springtails), and 51 mammals are catalogued [50]. Despite the city’s growth, this area has been preserved as it is important for the city’s water supply [51]. The BMBG receives annual visits of about 12,000 people, in addition to the population residing in the forest surroundings.

The MJSP (7°10′47″ S, 34°49′30″ W) is formed by Atlantic rainforest and tabuleiros fragments. The temperature variation is between 25 °C and 28 °C and relative humidity near 80% [52]. Unlike the BMBG, there is no information on plant, animal and fungi species. The MJSP is used for recreational activities, but there is no official estimate of the number of users.

### 2.2. Entomological Captures and Identification

HP light traps were installed at 176 locations divided proportionally between built areas (131 traps) and forest areas (45 traps) [53]. In the built areas traps were placed in backyards presenting favorable characteristics for the presence of sand flies, such as chicken coops, kennels, and decaying organic matter, or where cases of canine leishmaniasis had been reported (Figure 1B,C). In forest areas, traps were placed close to fallen tree trunks, exposed tree roots, animal burrows and decaying organic matter (Figure 1D,E). The captures were carried out simultaneously in built and forest areas from May 2019 to March 2020, August 2020 to February 2021, and March to July 2021. Discontinuation of the study occurred due SARS-CoV-2 pandemic restrictions. Each trap was placed for four consecutive days, corresponding to three nights of trapping [28], from 5 p.m. to 8 a.m. (totaling 45 h/trap).

The captures in the BMBG and MJSP were authorized by the Chico Mendes Institute for Biodiversity Conservation (ICMBio) under license nº 65740-1, and the research was registered in the National System for the Management of Genetic Heritage and Associated Traditional Knowledge (SisGen) under the code AC1D7D6.

For taxonomic preparation, specimens were processed according to the protocol of Vilela et al. (2018) [54]. Taxonomic identifications followed Galati (2021) [5], and genera abbreviations followed Marcondes (2007) [55].

### 2.3. Survey of Environmental Conditions

Environmental conditions such as humidity, temperature, precipitation, and wind speed were obtained from monitoring station (82798) of the Instituto Nacional de Meteorologia (INMET) of João Pessoa [56]. The average of the environmental conditions for the thirty days preceding the captures were used, except for precipitation, which was used as its accumulation in this period.

### 2.4. Data Analysis

Abundance was calculated as the total number of specimens captured, and richness was estimated as the total number of species sampled. The Rényi diversity profile and Jaccard similarity index were calculated to compare the diversity and similarity between the built and forest areas, respectively [57,58]. A Chao1 abundance-based diversity estimator was used to evaluate the true diversity of the sampling areas [59]. Species rarefaction curves were calculated to evaluate the effectiveness of sampling in each area [59]. A bioindicator species analysis was performed in order to describe the specificity of species as bioindicators of an environment [60].

To verify the correlation between sand fly abundance and environmental conditions, the abundance of the four most frequent species in each area was used if it was greater than ten specimens. Spearman’s correlation test was used to measure statistical dependence among species and environmental conditions (significance level = 5%; the correlation coefficient value rho ranges from −1 to 1 identifying, respectively, an inverse and direct correlation; the strength of this correlation can be characterized as null (rho = 0), weak (0.1 < rho < 0.3), moderate (0.4 < rho < 0.6), strong (0.7 < rho < 0.9), and perfect (rho = 1)) [61]. All analyses were performed using R software [62], using the Vegan Community Ecology package [63].

## 3. Results

A total of 2134 sand flies were captured; 33.2% females and 66.7% males. Nine genera and ten species (Table 1) were found: *Brumptomyia brumpti* (Larrousse, 1920), *Evandromyia* (*Aldamyia*) *evandroi* (Costa Lima and Antunes, 1936), *Lutzomyia longipalpis*, *Micropygomyia quinquefer* (Dyar, 1929), *Migonemyia migonei*, *Psathyromyia* (*Forattiniella*) *brasiliensis* (Costa Lima, 1932), *Psychodopygus claustrei* (Abonnenc, Léger and Fauran, 1979), *Sciopemyia sordellii* (Shannon and Del Ponte, 1927), and *Viannamyia furcata* (Mangabeira, 1941), also including one unidentified species in *Brumptomyia*. The following species are the first records to João Pessoa and to the state of Paraíba: *Br. brumpti*, *Mi. quinquefer*, *Pa. brasiliensis*, *Ps. claustrei*, *Sc. sordelli*, and *Vi. furcata*.

A total of 998 sand flies were captured in the forest areas. *Brumptomyia brumpti* was the most frequently captured species (574 specimens, 57.6%), followed by *Sc. sordellii* (171 specimens, 17.1%). *Brumptomyia brumpti*, *Ev. evandroi*, *Lu. longipalpis*, *Mg. migonei*, *Pa. brasiliensis*, *Ps. clautrei*, *Sc. sordellii*, and *Vi. furcata* were captured in the BMBG; *Ev. evandroi*, *Lu. longipalpis*, *Mi. quinquefer*, *Pa. brasiliensis*, and *Sc. sordelli* were captured in the MJSP. In the built areas 1136 sand flies were captured, with *Lu. longipalpis* being the most frequently captured species (978 specimens, 86.1%), followed by *Ev. evandroi* (149 specimens, 13.1%) (Table 2). The forest areas and built areas showed a similarity of 62%, with six species shared among them: *Lu. longipalpis*, *Br. brumpti*, *Sc. sordellii*, *Ev. evandroi*, *Pa. brasiliensis*, and *Brumptomyia* sp.; *Mg. migonei*, *Vi. furcata*, *Mi. quinquefer,* and *Ps. claustrei* were captured exclusively in the forest areas (Table 2).

Ten species were found in the forest areas, a similar value as expected for richness (Chao1 = 10, se. Chao1 = 0.000), while six species were found in the built areas (Chao1 = 6.5, se. Chao1 = 1.265). The species rarefaction curve of the forest areas reached asymptote (Figure 2A), differing from the built areas (Figure 2B).

The bioindicator species analysis verified five potential candidates: four associated with the forest areas (*Sc. sordellii*, *Br. brumpti*, *Vi. furcata*, and *Pa. brasiliensis*) and one associated with the built areas (*Lu. longipalpis*) (Table 3).

The Rényi diversity profile showed higher diversity in the forest areas compared to the built areas (Figure 3). The abundance of phlebotomines in the forest areas peaked in May 2019 with 213 specimens, while in the built areas this occurred in June 2019, with 158 individuals. Both peaks correspond to the rainy season, with precipitation of 161.6 mm and 216.5 mm, respectively. The correlations among species and environmental conditions (Table 4) showed that only *Lu. longipalpis* (built areas) and *Pa. brasiliensis* (forest areas) presented significant correlations with all the environmental conditions analyzed. Figure 4 shows the abundance of sand flies captured in the built and forest areas over the months.

No significant correlation with environmental conditions was observed in the forest area when analyzing the entire community of sand flies. However, in the built area, there was a positive and moderately significant correlation between the abundance of phlebotomine species with humidity (rho = 0.705, *p* value = 0.0005) and accumulated precipitation (rho = 0.636, *p* value = 0.0013).

## 4. Discussion

Deforestation impacts the diversity of sand flies in three different ways when comparing disturbed and undisturbed areas. First, the diversity in the disturbed areas is lower than in the undisturbed areas as reported by Oliveira et al. [64], Miranda et al. [40], and Pinheiro et al. [41]; second, there is no difference between disturbed and undisturbed areas as observed by Galati et al. [34], Lahouiti et al. [65], and Sanguinette et al. [66]; finally, diversity is higher in the disturbed areas than in the undisturbed areas as shown by Gonçalves et al. [67], Martinez et al. [68], Nascimento et al. [69], and Pereira et al. [70]. These differences probably occur due to the conservation status of the disturbed and undisturbed areas. Pinto Moraes et al. [42] demonstrated this by comparing a fragment of well-preserved forest with an urbanized zone, with a greater diversity of species in the forest area; however, by comparing a fragment of degraded forest, the urban area had more diversity. Ramos et al. richness [71] found that high human population density in combination with large forest cover was the most favorable scenario for maintaining sandfly diversity and richness.

The study found differences in the composition of sand flies between forest and built areas. The differences relate to the number of species per area and their frequency, with higher richness in forest areas. The results also suggest that the fragments analyzed do not show a high degree of anthropization. The predominance of *Br. brumpti* and *Sc. sordellii* in forest areas was similar to other studies [42,72,73], and these species are considered bioindicators in some municipalities of the states of Paraná (southern Brazil), Maranhão (northestern Brazil), and Amazonas (north Brazil).

The rainy season showed a higher abundance of sand flies, both in the built areas and in the forest areas. Similar results have been found in the Northeast, North, Central-West, and Southeast Brazilian regions [41,74,75,76,77]. The species found in the forest areas did not show significant correlation with the environmental conditions surveyed. Thus, other factors, such as availability of food sources for females, as well as inter- and intraspecific competition may be responsible for regulating these populations in forest areas, because the collection protocol found variations in the dynamics of the sand fly population throughout the study.

In urban regions, *Lu. longipalpis*—except regions with extreme climatic conditions—can be found throughout the year, and its seasonal pattern is almost always related to climatic conditions [78]. In the built area the abundance of *Lu. longipalpis* was correlated with the rainy season, a fact already reported to the city of João Pessoa [44]. Similar results have been reported in the Northeast, Southeast, and Central-West Brazilian regions [9,79,80,81].

Habitat changes modify the population response to abiotic conditions [68], which can cause a loss of diversity as observed among populations of the forest area and the built area. The built area showed a significant positive relationship with temperature and humidity, results already verified in urban areas in the Northeast and Central-West Brazilian regions [9,76,82]. Population increases in the immatures of *Lu. longipalpis* after rainfall have already been reported in urban areas [83].

The change in species composition among the forest areas and the built areas draws attention to vectors and possible vectors of leishmaniasis. *Lu. longipalpis* is a recognized vector of AVL [29,84]; *Mg. migonei* is a vector of American tegumentary leishmaniasis (ATL) [85] and AVL [30,39]; and *Ev. evandroi* was the only species found in an area of autochthonous transmission of AVL and ATL in Pernambuco. Therefore, its role as a vector of *Leishmania* in the region cannot be ruled out [86]. In the forest areas, *Ev. evandroi, Lu. longipalpis,* and *Mg. migonei* were found in low densities, being the fifth, sixth, and eighth most abundant species of the total of ten species. However, in the built areas, *Lu. longipalpis* is the dominant species, representing more than 86% of the total captured, followed by *Ev. evandroi*.

The presence of phlebotomine vectors of ATL, *Mg. migonei,* AVL, *Lu. longipalpis*, and *Mg. migonei* in the BMBG and MJSP is an important finding for the Health Secretariat of the municipality, given that the population using these spaces for recreational and ecotourism activities can become an accidental host in the enzootic cycle of AVL or ATL [87,88].

Changes in sand fly populations, with the prevalence of vector species, have been described previously. *Nyssomyia neivai* (Pinto, 1926) and *Pintomyia pessoai* (Coutinho and Barreto, 1940) were captured in abundance in built areas in Paraná, and they have also been identified as vectors of American cutaneous leishmaniasis (ACL) [72]. The prevalence of ACL and AVL vectors was also found in urban areas when compared to forest fragments in Mato Grosso do Sul (central-western Brazil) [89]. The environmental impacts of altering the course of a large river, such as the São Francisco River, also provided an increase of vector species *Lu. longipalpis* in residential areas [76].

The presence of *Lu. longipalpis* was first identified in 1973 in João Pessoa due to a AVL outbreak. At the time, 2520 people were examined, 70 of whom were infected, and the infection rate in dogs was estimated at 3.8%. Captures revealed a high specimen abundance in a region previously considered unsuitable for sand flies, due to its proximity to the ocean and constant wind [43]. *Evandromyia evandroi* has already been found in association with *Lu. longipalpis* in built areas in the metropolitan region of João Pessoa [90]. The results reaffirmed the connection of these two species with anthropized areas since both were more commonly found in built areas and the bioindicator species analysis showed that *Lu. longipalpis* is a bioindicator in the built areas in João Pessoa [29,91,92,93].

## 5. Conclusions

The study found that the built areas show differences in the richness and diversity of phlebotomine species in relation to the forest areas. In addition, the study showed that the conservation of forested areas, even urban fragments, favors the diversity of sand fly fauna. Furthermore, replacing forested areas with built areas has been beneficial for *Lu. longipalpis* (AVL vector).

The epidemiology of leishmaniasis is complex and involves studies which should observe vectors, reservoirs, parasites, socio–environmental reality, and the interactions between these components. Consequently, further studies are required to understand how the dynamics of AVL transmission occurs in João Pessoa. Based on this information, responsible government agencies can adopt more effective health measures to control leishmaniasis.

## Figures and Tables

**Figure 1 insects-13-01156-f001:**
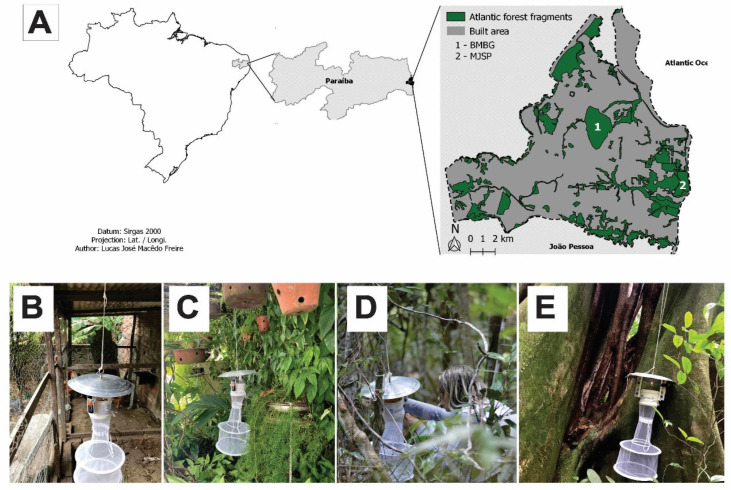
(**A**) Map of the municipality of João Pessoa showing the forest and built areas including the two forest fragments studied (points 1 and 2); (**B**) Point at MJSP; (**C**) Point at BMBG; (**D**,**E**) Points in the built area.

**Figure 2 insects-13-01156-f002:**
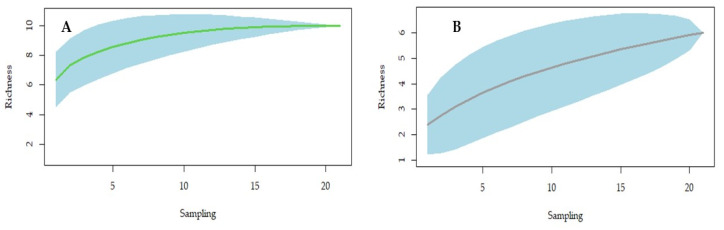
Rarefaction curves of sand flies in the forest areas (**A**) and the built areas (**B**) captured during the months May 2019 to March 2020, August 2020 to February 2021, and March to July 2021 in the city of João Pessoa.

**Figure 3 insects-13-01156-f003:**
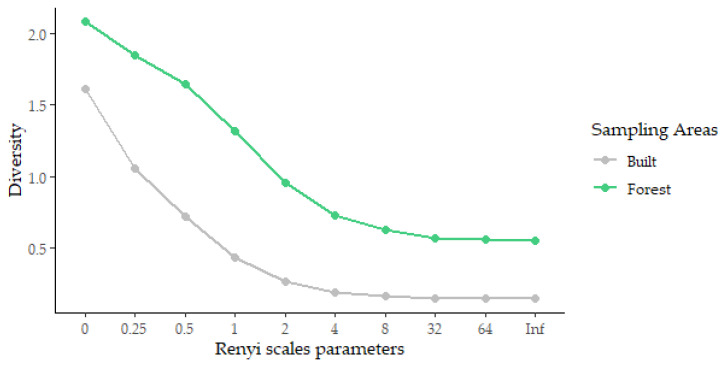
Rényi diversity profile of sand flies collected in the forest and urban areas from May 2019 to March 2020, August 2020 to February 2021, and March to July 2021 in the city of João Pessoa, state of Paraíba, Brazil.

**Figure 4 insects-13-01156-f004:**
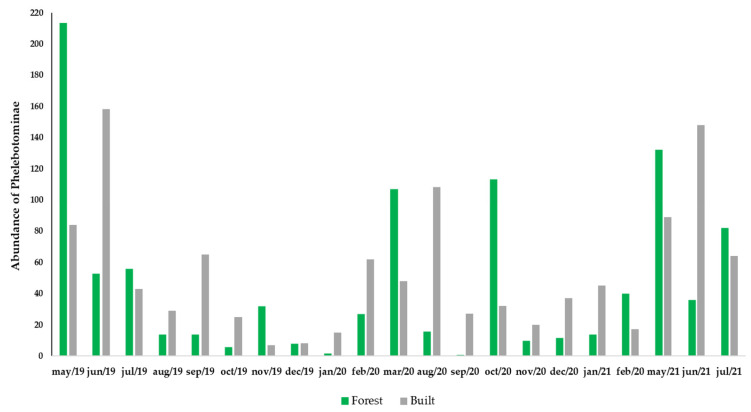
Abundance of sand flies in forest and built areas during capture months.

**Table 1 insects-13-01156-t001:** Species list, females and males captured in João Pessoa, Paraíba, Brazil.

Species	Females	Males	Total
*Lutzomyia longipalpis*	128	884	1012
*Brumptomyia brumpti*	287	289	576
*Evandromyia evandroi*	72	113	187
*Sciopemyia sordellii*	125	47	172
*Psathyromyia brasiliensis*	37	68	105
*Viannamyia furcata*	57	6	63
*Psychodopygus claustrei*	0	9	9
*Migonemyia migonei*	2	2	4
*Brumptomyia* sp.	4	0	4
*Micropygomyia quinquefer*	2	0	2
Total	714	1424	2134

**Table 2 insects-13-01156-t002:** Distribution of the sand flies captured in the forest areas and in the built areas in the city of João Pessoa, state of Paraíba, Brazil.

Species	Forest Areas	% Forest	Built Areas	% Built Areas	Total
*Lutzomyia longipalpis*	34	3.4%	978	86.1%	1012
*Brumptomyia brumpti*	574	57.5%	2	0.2%	576
*Evandromyia evandroi*	38	3.8%	149	13.1%	187
*Sciopemyia sordellii*	171	17.2%	1	0.1%	172
*Psathyromyia brasiliensis*	100	10.0%	5	0.4%	105
*Viannamyia furcata*	63	6.3%	0	0	63
*Psychodopygus claustrei*	9	0.9%	0	0	9
*Migonemyia migonei*	4	0.4%	0	0	4
*Micropygomyia quinquefer*	2	0.2%	0	0	2
*Brumptomyia* sp.	3	0.3%	1	0.1%	4
Total	998	100%	1136	100%	2134

**Table 3 insects-13-01156-t003:** Bioindicator species analysis of sand flies in the forest areas and in the built areas of João Pessoa, Paraíba, Brazil.

Species	Area	%	*p* Value
*Lutzomyia longipalpis*	Built	97%	0.005
*Sciopemyia sordellii*	Forest	77%	0.005
*Brumptomyia brumpti*	Forest	74%	0.005
*Viannamyia furcata*	Forest	62%	0.005
*Psathyromyia brasiliensis*	Forest	56%	0.010

**Table 4 insects-13-01156-t004:** Spearman’s correlation results between the most frequent species in the forest areas and built areas with environmental conditions, of João Pessoa, Paraíba, Brazil.

Species	Temperature		Humidity		Precipitation		Wind Speed	
	rho	*p*	rho	*p*	rho	*p*	rho	*p*
*Lutzomyia longipalpis* (built)	−0.41	0.000 *	0.80	0.000 *	0.80	0.000 *	0.77	0.000 *
*Evandromyia evandroi* (built)	0.08	0.729	0.18	0.430	0.18	0.000 *	0.10	0.649
*Brumptomyia brumpti* (forest)	0.04	0.838	0.09	0.704	0.09	0.704	0.23	0.312
*Sciopemyia sordellii* (forest)	0.26	0.249	0.08	0.704	−0.18	0.416	−0.20	0.372
*Psathyromyia brasiliensis* (forest)	−0.58	0.005 *	0.71	0.000 *	0.71	0.000 *	0.68	0.000 *
*Viannamyia furcata* (forest)	−0.47	0.031 *	0.10	0.644	0.10	0.644	0.12	0.584

Legend: * significantly *p* < 0.05.

## Data Availability

The data presented in the study are available in the article.

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
