# Peer review of "Ecological Aspects of the Phlebotominae Fauna (Diptera: Psychodidae) among Forest Fragments and Built Areas in an Endemic Area of American Visceral Leishmaniasis in João Pessoa, Paraíba, Brazil"

_insects, 2022, doi:10.3390/insects13121156_

Round 1
Reviewer 1 Report
In the item Entomological captures and identification, it is mentioned that the captures were carried out in 176 locations and proportionally divided between the studied areas (built-up area and forest fragments). However, data on the disposition, distribution and systematization of these captures are lacking, making it difficult to interpret the results. More details are also needed in the description of the capture sites, to facilitate the understanding of the proposed methodology. In addition, ecological data on forest fragments and built-up areas are lacking. What were the criteria chosen for these environments that characterized the choice of these capture sites. Another point that may reflect on the understanding and result of the composition of the species recorded in the studied areas, may be due to the discontinuity of captures over time, as it is known that in order to compare the fauna of certain species in ecologically distinct environments, it is necessary that the captures are systematized and made simultaneously during the same period of time. Aiming at a better understanding of the methodology used in the article, the construction of a graph is encouraged, taking into account the number of sandflies captured monthly in the two study areas, in the same period of time. Regarding the results obtained, it seems obvious to me that there is a greater diversity of species in forest areas when compared to built-up areas and that the species Lutzomyia longipalpis, the vector of visceral leishmaniasis, is adapted to the anthropic environment. These results are already well known and publicized in relation to the ecology of sandflies in Brazil.
Reviewer 2 Report
Authors provide an interesting work on sandflies population in Brazilian forest and urban envirnment. I have suggested few minor revisions.
Review
Title: “Ecological aspects of the Phlebotominae fauna (Diptera: Psychodidae) among forest fragments and built areas in an endemic area of American Visceral Leishmaniasis in João Pessoa, Paraíba, Brazil”
Introduction
Lines 41-43. Consider rephrasing the sentence. Leishmaniasis is not a clinical manifestations, but they include different symptoms, depending on the clinical manifestation, Visceral, Cutaneous, Muco-cutaneous Leishmaniasis.
Discussion
Lines 269-271. Consider rephrasing the sentence. “… because we found variations in the dynamics of the sand fly population throughout the study”.
Line 300. Consider replacing the word “impacts”. I didn’t understand what do you mean with impacts. May you describe the events, which influence sandflies population dynamics, more in detail?
Line 302. Replace LVA with AVL.
Line 305. Consider replacing the word “hitherto”.
Round 2
Reviewer 1 Report
Considering that the suggestions proposed in the first version, were accepted by the authors, I believe that the article can be accepted for publication.